# Few keystone plant genera support the majority of Lepidoptera species

Desiree L. Narango [1,2 ✉], Douglas W. Tallamy [1 ✉] & Kimberley J. Shropshire [1]

Functional food webs are essential for the successful conservation of ecological communities, and in terrestrial systems, food webs are built on a foundation of coevolved interactions between plants and their consumers. Here, we collate published data on host plant ranges and associated host plant-Lepidoptera interactions from across the contiguous United States and demonstrate that among ecosystems, distributions of plant-herbivore interactions are consistently skewed, with a small percentage of plant genera supporting the majority of Lepidoptera. Plant identities critical for retaining interaction diversity are similar and independent of geography. Given the importance of Lepidoptera to food webs and ecosystem function, efficient and effective restoration of degraded landscapes depends on the inclusion of such 'keystone' plants.

[1] Department of Entomology and Wildlife Ecology, University of Delaware, Newark, DE 19716, USA. [2] Present address: Department of Biology, University of Massachusetts, Amherst, MA 1002, USA. ✉email: dnarango@gmail.com; dtallamy@udel.edu

Several recent high-profile reports documenting a precipitous global decline in some insect populations[1–3] (reviewed in refs. [4–7], but see ref. [8]) have alarmed ecologists because of the primacy of insects in ecosystem services such as pollination and nutrient cycling as well as their role in food web complexity and stability and the maintenance of bird, reptile, amphibian, and mammal diversity[9–11]. Although some species may be stable or increasing[8], apparent reductions in any Lepidoptera population are particularly concerning given the importance of these insects in terrestrial food webs; caterpillars transfer more energy from plants to other animals than all other herbivores combined[12]. Loss of habitat, industrial farming, and pesticides are cited as primary causes of insect declines; as well as the pervasive creation of novel ecosystems[13,14] and their interference with specialized relations between insect herbivores and their host plants[15].

It is well-established that most insect herbivores exploit plants through specialized adaptations[16,17]. A wide array of phytochemical and physical defenses unique to each of thousands of plant lineages have forced most insect herbivores to develop adaptions that circumvent defenses specific to just a few of the available plant lineages[18]. Host plant specialization is most widely developed among mandibulate (chewing) insects such as caterpillars (the larvae of Lepidoptera moths and butterflies as well as Symphyta sawflies) because their feeding mode typically makes it impossible to avoid exposure to toxic or deterrent phytochemicals without specific physiological adaptations[16]. Thus, conserving caterpillar species necessarily means conserving the host plants to which they are physiologically and behaviorally tied. Which plant lineages host the most species of caterpillars, however, has never been formally quantified.

We have conducted a large-scale survey of Lepidoptera host plant records throughout the contiguous United States to search for patterns in host plant use that will facilitate insect conservation efforts aimed towards preserving species richness (Methods, Fig. 1). We focused our survey on Lepidoptera for two reasons: host records for this order of insects are more readily available and complete than for other insect taxa[19]; and Lepidoptera are inordinately important prey for insectivores[20–23], and perform ecosystem services that are ecologically, economically, and culturally important to people[9–11].

Our survey has revealed a previously unrecognized yet striking pattern: a small percentage of the plant lineages within a region support larval development in the vast majority of resident Lepidoptera (Fig. 1, Supplementary Tables 1 and 3). We call such hyper-productive plants "keystone genera" following the original reasoning of Paine[24]; as in Roman arches, keystone genera are unique components of local food webs essential to the participation of most other taxa in those food webs[25]. Without one or more keystone plants supplying energy to a food web, the web is predicted to collapse. Thus, to improve ecological function in degraded and cultivated ecosystems, plants that disproportionately support biodiversity must be included.

## Results

**Distributions of host plant-Lepidoptera interactions.** Across nearly all counties sampled, we found that distributions of host plants and larval Lepidoptera were highly skewed. Compared to other distributions, almost all county-level datasets best fit a Gamma distribution, such that the majority of plant genera support relatively few caterpillar species, while relatively few genera support many species (Fig. 1, Supplementary Table 2). For each county, we calculated the shape parameter α as a measure of distribution skew (i.e., how evenly Lepidoptera host relationships are distributed among available plant genera), and scale parameter θ as a measure of the steepness of slope (i.e., greater frequency of taxa that produce few to no Lepidoptera). For example, high values of α indicate that the shape of host plant use distribution is more evenly spread among available plant genera. High values of θ indicate a greater difference between plants that support many Lepidoptera species compared to plants that support few.

In general, we found that while parameters of α and θ varied slightly among ecoregions, relationships with plant diversity, Lepidoptera diversity, latitude, and county land area, effect sizes (based on unstandardized coefficients) were negligible and close to zero (Supplementary Figs. 1–4, Supplementary Table 1). Thus, while the number and identity of keystone genera is dependent on local diversity metrics and ecoregion, the overall pattern does not (Fig. 1). On average across the counties sampled, just 14% of the local plant genera support more than 90% of Lepidoptera diversity and thus serve as keystone plants throughout the United States (Supplementary Table 3). Moreover, the difference between the productivity of these genera and other genera is huge; keystone genera support orders of magnitude more Lepidoptera species than the majority of other local plant genera (Fig. 1).

**Network analysis.** To identify which plant genera disproportionately support Lepidoptera network structure at a national scale, we modified methods from Harvey et al.[26], and used a bipartite analysis on a binary network of host plant-Lepidoptera interactions (1 = interaction, 0 = no interaction). For this analysis, we calculated the following metrics for each plant in each county: (1) species richness, (2) extinction sensitivity and (3) network stability. Species richness is the total number of Lepidoptera species known to feed on a host plant. Extinction sensitivity is the total number of specialized Lepidoptera species that exclusively feed on a host plant and would be extirpated if that host plant were removed. Network stability is the relative change in overall network stability following the sequential removal (and replacement) of each basal host plant and all associated interactions with Lepidoptera using bootstrapping[26,27]. Following Suave et al.[27], stability is defined as the minimum number of intraspecific interactions necessary for a stable network where the smaller the value, the more resilient a network is to disturbance (see Suave et al.[27], Appendix 1 for more details). A plant's contribution to stability is thus the difference between the initial stability when all interactions are present, and the stability when a basal plant, and all subsequent interactions, are removed (refs. [26,27] for additional details on the quantitative analysis, and Methods for additional details on our modifications). The combined contributions of species richness, extinction sensitivity, and network stability to host plant–Lepidoptera interaction networks identify particular plants that will have a larger impact on network structure and thus can serve as conservation targets for land managers[26].

Our analysis revealed that a few genera were consistently identified as the top performers across the United States. Ten plant genera were identified as outliers due to their disproportionately high values relative to the mean across all county networks analyzed (mean = 0.14, Fig. 2, outliers are 1.5 × length of the interquartile range). The top 5 genera were *Quercus* ('Oaks', mean score: 0.79), *Salix* ('Willows', 0.55), *Prunus* ('Cherries, Plums, Peaches, etc.', 0.51), *Pinus* ('Pines', 0.46), and *Populus* ('Poplars, Aspens, and Cottonwoods', 0.44) (Fig. 2). Although some of these genera are already recognized as important components of ecosystems by some ecologists, their value may be underappreciated in terms of their contribution to food webs, as well as their importance in managed, cultivated landscapes, and our results demonstrating their importance is apparent at a macroecological scale.

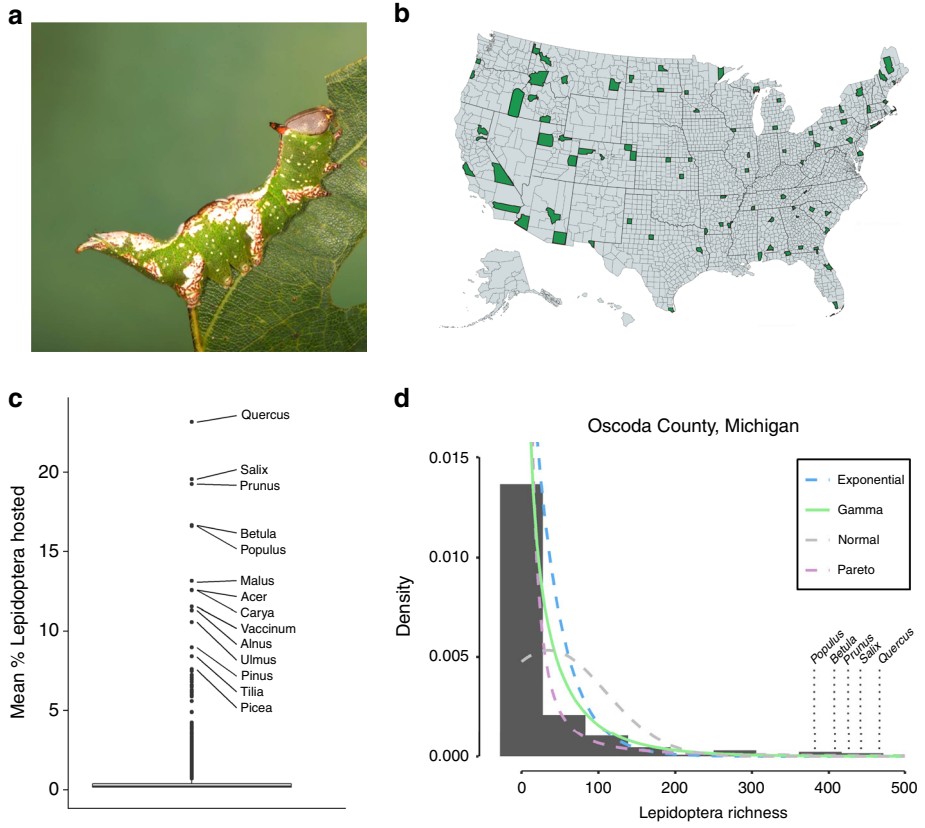

**Fig. 1 Across the United States, distributions of Host Plant-Lepidoptera interactions are skewed such that few plant genera support the majority of Lepidoptera species. a** White-blotched Heterocampa (*Heterocampa umbrata*) a Lepidopteran from the family Notodontidae. The majority of Lepidopteran caterpillars are constrained to feed on just a few plant genera or families from the total plant diversity available in the landscape (Photo by D. Tallamy); **b** The 83 United States counties (in green) used in the analyses (map made by KJS using mapchart.net); **c** Mean % Lepidoptera species hosted by each plant genus across all counties analyzed (*n* = 83). The top 15 genera that supported the most Lepidoptera species are labeled. The average percent Lepidoptera species across all plant genera was 0.36% ± 1.41% (range: 0–23%). The box plot shows the median (center line), first and third quartile (upper and lower hinges) and 1.5 * interquartile range (whiskers). Data from *n* = 1997 native plant genera. **d** Example histogram of Lepidoptera richness by plant genus with four candidate distributions for the 341 woody and herbaceous plant genera in Oscoda County, Michigan. Most (93%) of the 83 counties included in the dataset had distributions that were best fit by a gamma distribution (solid green line), such that a few plant genera supported high numbers of Lepidoptera species and many plant genera supported few to no Lepidoptera species.

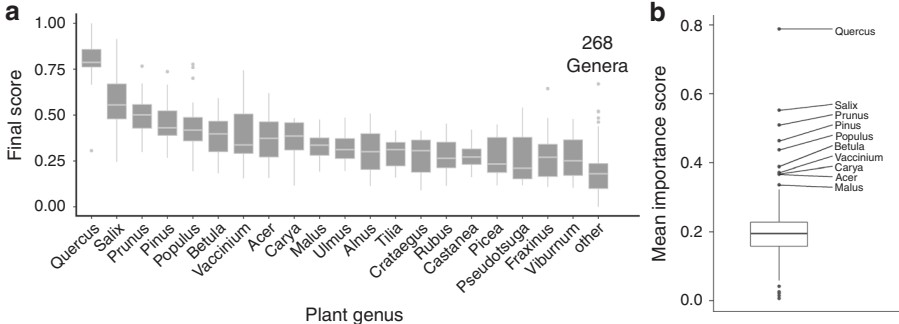

**Fig. 2 Some plant genera are disproportionately important for supporting host plant- Lepidoptera interaction networks. a** Mean importance scores for the top 20 plant genera (plants with the highest scores that are also present in three or more counties) and all other plants combined. Importance scores were calculated as the standardized rank score that combined (1) the number of Lepidoptera species hosted, (2) the number of specialized Lepidoptera species supported (species that only use one plant genus), and (3) the effect on network stability. **b** Mean importance scores for the 198 woody plant genera that support at least one Lepidoptera species included in the keystone plant network analysis. 10 plant genera were identified as outliers (>1.5* interquartile range). Plant genera not known to support any Lepidoptera species were excluded from this graph (*n* = 90). All box plots show the median (center line), first and third quartile (upper and lower hinges), and 1.5 * interquartile range (whiskers).

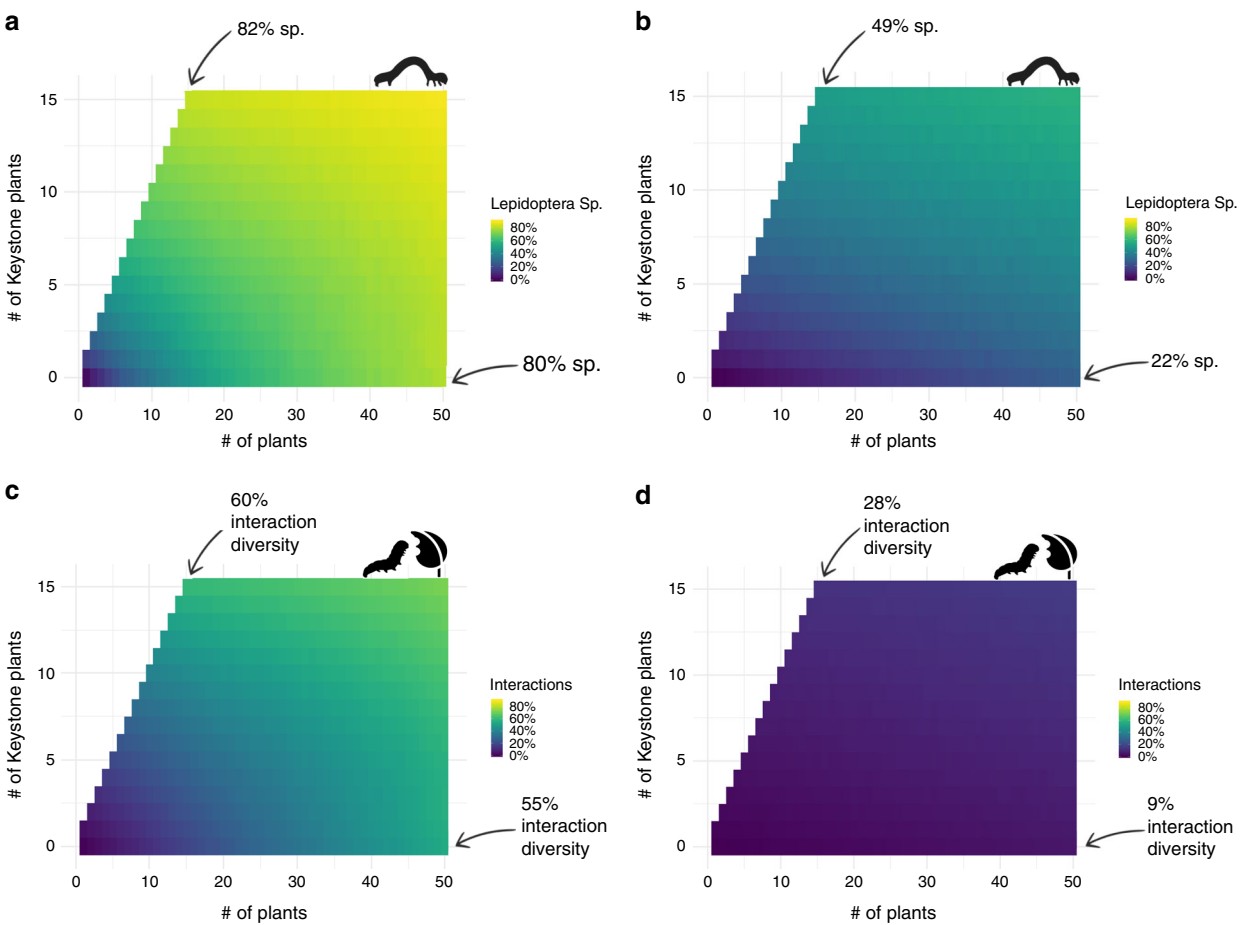

**Fig. 3 Including woody keystone plants increases the efficacy and efficiency of the restoration of Lepidoptera diversity.** Simulations of restoration scenarios where (**a**, **b**) Lepidoptera diversity and (**c**, **d**) interaction diversity as numbers of plant genera (*x*-axis) and keystone plant genera (*y*-axis) increase in woody and herbaceous plants. For comparisons across counties, percent species and percent interactions from total possible is used in lieu of raw numbers. For both woody and herbaceous plants, supported caterpillar species and interactions were higher with fewer plants when keystone plant species were included. In woody plants, more than twice the number of plant genera was required to achieve the species and interactions produced by keystone plants. In herbaceous plants, comparable richness and interactions were not reached even with a 3-fold increase in plant genera richness. Caterpillar icon in (**a**, **b**) from www.phylopic.com; caterpillar and leaf icons in (**c**, **d**) made by Freepik [https://www.flaticon.com/authors/freepik] from www.flaticon.com.

**Restoration simulations**. To demonstrate the utility of identifying keystone genera, we conducted a simulation of restoration scenarios demonstrating how Lepidoptera diversity and interaction diversity changes as both plant genus richness and keystone plant richness increases. To standardize across counties, we calculated the percent Lepidoptera species supported (from all possible phytophagous species) and percent interactions supported (from all possible plant-caterpillar interactions). Our simulation revealed that restored Lepidoptera and interaction diversity could be nearly doubled by the presence of just a few productive taxa for both woody (Fig. 3) and herbaceous plants (Fig. 4). For example, when 20 woody plants are chosen randomly, just 38% of Lepidoptera and 13% of interactions are supported; in contrast, when ½ of selected species are keystone plants, 71% of Lepidoptera and 40% of interactions are supported (Fig. 3). Differences were even more apparent for herbaceous plants, where plant selection for 20 plants without keystone species supported just 13% of potential species and 2% of potential interactions but with the inclusion of 10 keystone species, 42% of Lepidoptera were supported and 10% of interactions.

Although increases in genus richness may eventually achieve similar gains in Lepidoptera and interaction diversity, including keystone plants achieved the highest diversity with the fewest plants. For example, 82% of possible Lepidoptera species and 60% of interactions could be supported with 15 woody plants when keystone plants are intentionally included, whereas, 50 woody plant species were required to reach 80% of the Lepidoptera species and 55% of interactions when plants were chosen randomly (Fig. 3). For herbaceous plants, even a threefold increase in plant richness was unable to capture the diversity of Lepidoptera species and interactions supported by fewer keystone plants (Fig. 3). Thus, restoration actions that populate landscapes with native plants, but without members of keystone genera, are performing ineffectively and are unlikely to support similar diversity of local Lepidoptera species compared to landscapes that included keystone genera.

**Discussion**
Recent studies measuring how various plant assemblages impact wildlife populations view plants in novel ecosystems in terms of a native-nonnative (i.e., 'exotic', or 'introduced') dichotomy[28–33]. Our analyses suggest that this approach is functionally simplistic: native plants, even within biomes, are not all equivalent in terms of their contributions of energy to food webs. Without recognizing the outsized effect of keystone plants on the energy flow through food webs, plant selection for landscape projects that is

based only on commercial availability, aesthetic criteria, or dual use such as lumber or fruit production[34,35], will be unable to support the richness and diversity of species necessary for robust ecosystem function, even if selections are confined to native flora.

Historically, conservation has been a reaction to single species in crisis. However, no species exists in isolation of other species and to achieve conservation goals, the focus must shift to preserving interaction diversity[25,26,36–38]. Identifying, conserving, and/or restoring keystone plants that form the hubs of interactions in complex food webs will help prioritize the selection of plants for insect conservation efforts and thus the conservation of insectivores in higher trophic levels.

The pattern that has emerged from our host plant analyses is suggestive and inspires new research questions, including: (1) Is the diversity and abundance of insectivores in higher trophic levels that is predicted by Lepidoptera host plant records actually a function of the presence and abundance of keystone plant genera? (2) Does the pattern we observed in North America host records occur on other continents? (3) Do keystone plants occur within herbivorous insect taxa other than Lepidoptera? (4) How much keystone plant biomass is required to drive local food webs?

Several factors might contribute to the skewed Lepidoptera productivity exhibited by keystone genera including genus size, geologic age, phylogenetic isolation, geographic distribution, relative apparency in the landscape, and mode of chemical defense[39–43]. Regardless of their genesis, however, the implications of keystone genera for conservation and restoration are profound wherever plants are managed in North America or, if they prove to be a general phenomenon, globally. Landscapes that do not include keystone genera may produce on average half the number of species of Lepidoptera which is likely to result in lower and more erratic caterpillar abundance and thus impede the conservation of ecologically and culturally important insectivores such as birds. For example, in our dataset, 138 native woody genera and 860 native herbaceous genera are currently not known to support any Lepidoptera diversity in the United States at all. Although future sampling will undoubtedly yield previously undocumented interactions in many of these plant genera, particularly for rare or uncommon Lepidoptera or plant species, they are unlikely to be numerous enough to substantially change patterns of distribution we report here. The recognition of the role of keystone genera in local food webs should discourage the profligate use of introduced plants as well as unproductive native plants in reforestation, agroforestry, urban forestry, horticultural landscaping, land reclamation, natural area buffers, pollinator habitats, and carbon offset plantings without regard to their ability to function within novel ecosystems.

Here, we adapted an analytical framework from Harvey et al.[26] to identify plants that could be disproportionately useful for restoration, and further demonstrate how network structure can inform conservation application for land managers. While the results are useful for plant selection generally, the approach used here can also be applied to finer-resolution data to provide inference at local scales (For an example, see Supplementary Figs. 5–6 for a comparison with a field-collected dataset). An important caveat is that our results primarily inform the preservation of species richness; in some cases, assessing the abundance and/or biomass may be comparably important for maintaining specific Lepidoptera populations and or consumers that rely on Lepidopteran prey. For example, when a caterpillar species uses multiple host plant genera, the species may be more or less abundant on some host plants over others due to differences in host plant chemistries, foliar nutrition, competitive exclusion, or regional specialization[44,45]. Thus, for restoration goals aimed toward specific plant or Lepidoptera species, rare or uncommon taxa, or particular tracts of land, locally relevant host plant use that incorporates abundance or biomass may be necessary. However, at present, abundance data of caterpillars on host plants do not exist at the national scale or across full plant communities as considered here. Moreover, insect abundance can fluctuate wildly over time, space, and with the collection method, which makes meaningful comparisons based on abundance over large scales challenging. Although our inference is on Lepidopteran species supported, richness is often strongly correlated with other metrics of diversity like abundance and biomass across taxa[46,47]. Thus, the preservation of species richness will likely also translate into meaningful preservation of abundance, biomass, and ecological service.

We do not suggest that the keystone plant pattern observed in our survey is a call to reduce plant diversity in managed ecosystems. On the contrary, diverse plantings are more likely to contain one or more keystone plants by chance alone and increasing plant diversity also increases the diversity of supported arthropods. Our results reveal that increasing plant diversity with the intentional inclusion of keystone plants is the most efficient and successful approach to support Lepidopteran diversity. Moreover, fueling insect-based food webs is only one essential ecosystem service that plants provide. At a minimum, plants must also support a diverse community of pollinators, manage watersheds, and sequester carbon. Because plants vary so widely in their ability to accomplish these ecological goals, diverse plantings are more likely to produce such services better than simplified plantings. Nevertheless, in managed or restored ecosystems, where plant community composition is predetermined, our study suggests that keystone plants are a necessary if not sufficient criterion for the creation of robust and diverse food webs.

## Methods
**Data collection**. This study uses a compiled data set that includes 12,072 native Lepidoptera species, 2079 native plant genera, and 24,037 different host plant-native Lepidoptera interactions from across the contiguous United States. From these data we extracted the plant information of 83 counties and a Lepidoptera list for the corresponding 25 states. The full dataset will be publicly available in a forthcoming data manuscript.

**Lepidoptera range and host plant data**. The Lepidoptera species data were compiled from historic citable sources (Supplementary Data 6) of range and host plant records. We originally compiled a similar list for the Mid-Atlantic region[29]. This dataset was updated to include more states and counties to include on the National Wildlife Federation website[48]. Non-plant host records (e.g., detriphagous, algae, fungi, lichen, and insect predators) are included, as well as Lepidoptera without known host plant associations, but not considered in this analysis. Plant ranges are to the county, Lepidoptera ranges are to state, and host plant-Lepidoptera records are relationships between a plant genus and Lepidoptera species. Plant genus was included as the unit of interaction because data on Lepidoptera- host plant associations are most accurate and available at the genus level. Although more specific data are occasionally available (e.g., Lepidoptera records to plant species), we limited our analysis to the genus scale in order to make equitable inferences across the geographic and ecological scope of this analysis.

**Plant distribution data**. The current list uses the Biota of North America Program[49] (BONAP) as its major source for plant nomenclature. We used the BONAP database as our source for plant distributions because it specifies North American plant ranges that currently occur beyond their historic native range due to anthropogenic and natural expansion (e.g. Osage orange, *Maclura pomifera*).

Using the BONAP, a county-level survey was made for each state used in this study. Every county within those states was surveyed and BONAP records include records from adjacent counties. We classified plants into three categories; native, non-native, and adventive. A plant species is classified as adventive if it is native to North America but not in that specified region. Each plant genus was reviewed individually in each state. Genus records that fell entirely in one category resulted in all county records being designated that category. Any plant genus that had species that fell in two or more categories was examined county by county, with adjacent records being noted but not included. The state records for each plant genus are labeled in various combinations of the three categories. County-level data designate a genus either containing native records, or only non-native. For our study, we focused only on native plants, excluding non-native and adventive

records (except for parameterizing probabilities of host-plant switching, see below for details).

Eighty-three counties in 25 states (Alabama, Arizona, Arkansas, California, Colorado, Delaware, Florida, Georgia, Idaho, Illinois, Kansas, Maine, Massachusetts, Michigan, Minnesota, Montana, New York, North Dakota, Ohio, Oregon, Pennsylvania, South Carolina, Tennessee, Texas, and Utah) were examined. At least three counties in each state (except Delaware) were used. Two counties from each state were selected from dissimilar ecoregions within each state. Ecoregions were determined using the Commission for Environmental Cooperation's Ecological Regions of North America map[50]. For county selection we used the level 2 designation of terrestrial ecoregions (50 separate categories); however, for subsequent analyses, we used the level 1 designation (15 categories). As much as possible, counties that bordered other counties within an ecoregion were used to alleviate the issue of BONAP including records from adjoining states. At least one more county was added to meet the parameters of the latitude study and to fill out ecoregions. Up to five counties were used in some states. While the majority of counties were chosen without criteria beyond ecoregion status, Chase County KS was chosen based on its presence in the 'South-central Semi-Arid Prairies' ecoregion, as well as high natural grassland cover relative to other agriculturally dominated Kansas counties.

**County data**. A series of counties were selected along three primary latitude bands (Latitudes 46, 40, and 34). The latitude and longitude of each county were determined by the county seat using Google Earth. In most cases the county seat was centrally located within the state. A few county seats are not centrally located, most notably Monroe County in Florida where the county seat is Key West. We determined the land area (in km², excluding inland, coastal, Great Lakes, and territorial sea water) for each county using information from the US Census (https://www.census.gov/quickfacts/fact/note/US/LND110210). Counties varied from 415 to 26,368 km² with an average of 4519 ± 5598 SD.

**Lepidoptera-host plant data**. The host plant records for each Lepidoptera species include all known literature records. Not all host plant-Lepidoptera associations occur in every county, state or even the USA due to differences in plant distributions. Thus, we filtered the Lepidoptera list from each state to exclude any Lepidoptera species whose host plant did not occur in the selected county. The final dataset per county includes (1) all plant genera known to occur in the county, (2) all Lepidoptera known to use at least one plant genus that occurs in the county and (3) all host plants used by Lepidoptera that could potentially occur in the county.

**Statistical methods**. All analyses were conducted using program R, Version 3.5.1[51].

**Distribution analysis**. We first determined what distribution best fit our data in order to derive parameters that could be compared among counties. We used the package 'goft'[52] to conduct goodness of fit tests for the Exponential, Gamma, and Pareto distributions on each county separately. Functions in the 'goft' package use parametric bootstrap tests for the null hypothesis that a distribution fits a tested distribution. We tested each county ($n = 83$) and each distribution type ($n = 3$) separately. Using the distribution that best fit our datasets, we then used the function 'fitdistr' from the 'MASS' package[53] to use maximum likelihood fitting to obtain parameters (e.g., shape $\alpha$ and scale $\theta$) for the Plant-Lepidoptera distributions for each county separately.

We then tested for differences in the distribution of caterpillar richness among plants by county-level diversity and location metrics. For each county, we compared the $\alpha$ and $\theta$ of the distribution with county plant richness, Lepidoptera richness, ecoregion, latitude, and county land area. To test whether $\alpha$ or $\theta$ varied by ecoregion we used an analysis of variance (ANOVA) with Tukey's post hoc multiple comparisons of means. To test whether $\alpha$ or $\theta$ changed with increasing plant richness, Lepidoptera richness, or land area, we used linear regression. Based on scatter plots of the data, no nonlinear relationships were necessary.

**Network analysis**. To determine conservation targets, we identified keystone plant species[24,25] using methods from Harvey et al.[26]. To perform this analysis, we used binary networks of host-plant caterpillar interactions. Ecological networks are ideal to determine cascading extinction rates of specialists following host plant loss[54]. For this analysis and our following simulation, we chose one representative county dataset from each state from our 83 available counties (25 counties total). Our method to identify target keystone species at a national scale consists of three steps.

**Species richness**. On a per-county basis, we first identified how many Lepidoptera species are recorded in the literature as using each plant genus for growth and reproduction.

**Extinction sensitivity**. We also determined the 'extinction sensitivity' of each plant; in other words, how many Lepidoptera species are at risk of extirpation with the loss of a host plant? In our context, the sensitivity means "specialization", caterpillars that use only one genus of the plant are considered especially sensitive

to the removal of that plant. We modified the "nb.extinct" function from Harvey et al.[26] so that we could calculate the number of extinct herbivores following the removal of a plant. This function was repeated for each county to acquire a number of species that were specialists to each host plant.

**Network stability**. Then we used a network-based approach to assess the effect of each plant genus on network stability. We used a binary matrix where each record of a caterpillar on a host plant indicates the existence of an interaction. The results given from a binary interaction network are correlated with those from a network weighted by abundance[55]. Here, the community stability index represents the minimum interactions required for the system to be stable where smaller values are the most stable, i.e., the more resilient a network is to disturbance, and large values are the most unstable. We calculated the stability index as the real part of the dominant eigenvalue of a Jacobian matrix following methods from Sauve et al.[27] (see Appendix 1 in ref. [27] for definition and details). To acquire baseline stability, we conducted 175 iterations and took the median value. We determined that 175 was the minimum number of iterations needed to acquire a stability value ± 0.001 resolution using simulations on test datasets. For this analysis we only considered woody plants in our analysis because (1) woody plants tend to host the most diverse caterpillar communities[29] and (2) computation time to include all plant genera was prohibitive.

To quantify the effect of each plant on total network stability, we reran the analysis, iteratively removing each plant genus and then recalculating the stability[26]. Then we subtracted the new stability values from baseline stability to find the median change when each plant was removed where negative values indicate reductions in network stability and positive values indicate increases. We then multiplied the stability value by −1 so that increases in this metric indicated an increase in a plant's importance to stability to make this value comparable in direction to network structure and extinction sensitivity (i.e., increases in Lepidoptera diversity and # of specialist species).

**Standardizing results across counties**. For each analysis (step 1–3) we scaled our final values from 0–1 using this equation for each plant in each county separately:

$$\frac{x + |\text{minimum } x|}{\text{maximum } x + |\text{minimum } x|}. \tag{1}$$

We then took the mean of our three values to obtain a final 'score' for each plant genus per county. Finally, we identified which plant genera had the highest values over all the counties by plotting the means for each plant genera ($n = 288$) and assessing outliers (values that were 1.5× the interquartile range).

**Field-based host plant-caterpillar interaction data**

*Field sampling*. To compare the results from the network analysis on literature-based data collection with results from field-based data collection, we used caterpillar interactions recorded from native host plants from Richard et al.[56] (hereafter: Mid-Atlantic dataset). Caterpillar surveys were conducted in 2011 within 8 hedgerows in New Castle County, DE and Cecil County, MD. This dataset contains plants surveyed in both native- (>95% native plant biomass, $n = 4$) and nonnative-dominated (>75% nonnative plant biomass, $n = 4$) hedgerows. Sites were all located within Mid-atlantic decidous piedmont forest, and were separated by at least 100 m.

In June–July, an observer walked a 100 m transect on days in which foliage was not wet to collect caterpillars in each site. Observers sampled all caterpillars using the total search approach[57] to methodically inspect leaves, twigs, and branches of all woody plants within a 2-m3 area along the transect. Each search was conducted for 5 m every 2 m along the 100 m transect. In total, each hedgerow treatment was searched for a total of 1000 min in both June and July. All caterpillars were identified to species or morphospecies using Wagner[57], Wagner et al.[58], and various web sources. Caterpillars that could not be identified in the field were measured and then brought to the lab to be reared to adulthood for later identification using the literature and the University of Delaware Insect Reference collection.

*Data management and analysis*. Because our network analysis was based on native woody plant genera, we excluded all non-native plant genera in the Mid-Atlantic dataset. We calculated the number of times each plant was searched and excluded all plant species that were searched <10 sampling occasions. That left us with caterpillar abundance data for 18 genera: *Acer (A. rubrum* and *A. saccharum), Carpinus caroliniana, Carya sp., Cornus (C. alternifolia, C. florida*, and *C. race-mosa), Diospyros virginiana, Fagus grandifolia, Fraxinus americana, Juglans nigra, Lindera benzoin, Liquidambar styraciflua, Liriodendron tulipifera, Platanus occi-dentalis, Prunus (P. americana* and *P. serotina), Quercus (Q. alba, Q. montana, Q. rubra, Q. velutina, Q. palustris, Q. coccinea, Q. phellos*, and *Q. imbricaria), Rhus (R. glabra* and *R. copallinum), Sassafras albidum, Ulmus (U. americana* and *U. rubra), and Viburnum (V. dentatum* and *V. prunifolium)*.

Prior to analysis, we filtered the Mid-Atlantic dataset to exclude all caterpillar species that weren't definitively identified to species. Excluded species were primarily leaf miners, tiers, folders and webbers. Included individuals were composed of species from 19 separate families.

We summarised whether the 58 identified caterpillar species were observed feeding on the 18 host plants and converted the data into a binary matrix. For abundance, we standardized search efforts across the plant species by summing the total number of surveys and calculated an abundance per 10 samples. We completed the network analysis on the field-collected interaction dataset using 10,000 iterations. We compared the values from the Mid-Atlantic dataset (Field-derived scores) with that derived from the host plant data (Literature-derived scores) using a Pearson's correlation test using the function 'cor.test'[51] and report the R-statistic.

### Restoration simulations

*Simulation parameters*. To determine the applicability of our results, we simulated the management actions of restoration efforts involving 1–50 plant genera and 0–15 keystone plants (i.e. plants that support disproportionately high Lepidoptera richness) to demonstrate how supported Lepidoptera and interaction diversity changed when keystone plants were intentionally included in the landscape or not. To do this, we simulated random plant choices from the list of available plant genera from each of the randomly selected counties from 25 states and calculated the total Lepidoptera species richness supported (excluding duplicate species supported by >1 chosen host plant) and total interaction richness supported (i.e. all interactions between a plant and a caterpillar consumer).

*Host-plant switching*. We also included the potential for host plant switching by including a probability of using plants not recorded in the host plant literature. We calculated the probability of random host plant switching as:

$$P_{ic}(hostplantshift) = \frac{E_{ic}}{N_c - H_{ic}} * N, \qquad (2)$$

where $P_{ic}$ is the probability of host plant shifting by Lepidoptera species $i$ in county $c$, $E_{ic}$ is the proportion of non-native plants used by Lepidoptera species $i$ in county $c$, $N_c$ is the number of native host plants in county c, $H_{ic}$ is the number of total native host plants used by Lepidoptera species $i$ in county $c$, and $N$ is the total number of plants included in the simulation (from 1 to 50). This equation gives the probability that Lepidoptera species $i$ in county $c$ shifted to at least one of $N$ plants used in the simulation. Using this probability, we used the sample function in R to predict whether any of the possible Lepidoptera species were included or not in each iteration and added each unique species to those included from known hosts.

*Simulation output*. For each county, we iterated these scenarios over 100 iterations with random draws of plant genera and keystone genera. We chose 100 iterations for each county to accurately estimate means while maintaining computational efficiency. To standardize across counties, we calculated the percent of Lepidoptera supported out of all potential phytophagous Lepidoptera (for woody plants and herbaceous plants separately) and percent interactions in each iteration. For each county and iteration, we plotted the median value. Simulations were run for woody and herbaceous plants separately.

### Reporting summary

Further information on research design is available in the Nature Research Reporting Summary linked to this article.

## Data availability

Plant distribution data for this study was collected from The Biota of North America Program (BONAP): http://www.bonap.net/tdc (2014, and continuously updated). The literature and databases used to collect information on Lepidoptera distributions and host plant-Lepidoptera interactions are included as a bibliography in Supplementary Data 6. The final dataset analyzed for this study, which includes the number of Lepidoptera species by host plant genus for each county, as well as county-level information, is included as Supplementary Data 1–2. The full dataset of all Host plant genus – Lepidoptera species interactions for each state will be published in a forthcoming data paper and can be available upon reasonable request.

## Code availability

The code to conduct the analyses is included in Supplementary Data 3–5.

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

## Acknowledgements

The scope of this project would not have been possible without the dedicated work and passion of over a century of naturalists, entomologists, and botanists recording observations of plants, Lepidoptera and host plant-caterpillar interactions. The authors thank Chelsea Cox for her contributions to compiling the data and M.T. Hallworth for helpful conversations about the analyses. Funding to D.W.T. from the National Wildlife Federation/USDA Forest Service. Funding to D.L.N. from the University of Delaware Graduate School Fellowship.

## Author contributions

Conceptualization, D.L.N., D.W.T., and K.J.S.; Methodology, D.L.N., K.J.S and D.W.T.; Investigation, D.L.N., K.J.S and D.W.T.; Formal Analysis, D.L.N.; Writing—Original Draft, D.L.N. and D.W.T.; Writing—Review & Editing, D.L.N., K.J.S., and D.W.T.; Funding Acquisition, D.W.T.

## Competing interests
The authors declare no competing interests.
