## [Peer Review File · Nature Communications]

Reviewers' Comments:

Reviewer #1:

Remarks to the Author:

The manuscript titled "Identifying Keystone Plants; The Backbone of Insect-Based Food Webs" describes a data synthesis effort reporting which genera of host plants support disproportionate number of lepidopteran herbivores. The authors posit that conservation of these "keystone genera" would be an important step toward stabilizing entire food webs. The authors assembled and present a novel data set in a manuscript that elegantly moves from theory to practical conservation measures. Although others have monitored interaction patterns in nature, documented loss of food resources, or highlighted the importance of interaction diversity as a stabilizing force in food webs, few studies have shown how conservation plans could include interaction diversity in a practical way. The manuscript is clearly written, and will likely be of broad interest to theoreticians and conservation biologists alike. Modification and additional information regarding three main and a few specific points would improve the manuscript:

Main comments:

1) The narrative of the introduction paragraph 1 lines 12-20 seems to "cherry-picked" as one of the best long-term monitoring studies of insects show increases and decreases of moths over time (Macgregor et al NEE, 2019), whereas some of the studies cited in this manuscript as evidence of unequivocal declines are commentaries that provide no data regarding declines (Eisenhauer et al Nat. Com 2019), or have highly controversial experimental designs (Hallmann et al 2017 PLoS ONE).

2) The key result of the manuscript is that a small percentage of plant genera support the majority of Lepidoptera. This is determined looking at presence and absence of lepidoptera on host plants, rather than abundance of lepidoptera. If abundant species are found on separate host plant species (sensu segregation-aggregation patterns reported in Calatayud et al 2019 Nature Ecology and Evolution) then a wider variety of hosts would need to be conserve total lepidopteran biomass, but a narrower range of hosts would be needed to conserve species richness. The influence of presence-absence vs abundance data on the outcome of the results should be qualitatively described as an important caveat, or quantitatively included in the simulations as a sensitivity analysis.

3) The authors indicate that the host plant associations are filtered because not all lepidopterans occur in all areas where their potential host plants occur. However, the choice for units for variables are not standardized metrics (i.e. county area) or carefully justified (ie. genus). Why were plant genera chosen as a unit of conservation? How does the number of species per genera influence their importance? How did the authors account for differences in sizes of counties? Looking at the map in Figure 1B there appears to be quite a big size range. Noting that the largest county in the US is 51,950 km² and the smallest in the continental US is 59 km². It is likely then that the network of potential interactions in larger or more heterogeneous counties end up with more interactions that are not actually realized.

Specific comments:

Line 53: multiple effect sizes but only one value of β ? The reader is not informed to what this parameter refers until the methods at the end. A brief description that parameter in the text here would improve ease of reading.

Figure S2 slopes were significant but effect sizes were near zero—How is effect size calculated?

Line 61-63- network analysis is a bit vague. Stability is not clearly defined—add more explanation in addition to the Harvey et al reference.

Line 173: The terms 'zone', 'ecoregion', and 'bioregion' are all used in the methods. If they refer to the same thing, they should be standardized, preferably consistent with the cited reference.

The cited reference to ecoregions describes a hierarchy of regional partitioning. Yet the application of the hierarchy, or which level is chosen is not clearly described in this manuscript.

Line 199: Why is productive specified?

Reviewer #2:

Remarks to the Author:

This interesting and important manuscript is based on a substantial trophic database that is focused on the host plants of lepidopteran larvae (as well as the consumers of these caterpillars). A clear pattern of skewed distribution of food plant genera show a small percentage of "keystone" plants, and the authors suggest that these plants are essential for restoration or for preventing continued declines of Lepidoptera, which are important pollinators and primary consumers.

The work is certainly important with broad scientific appeal and interest and it is worthy of publication in a high impact journal. While the methods, interpretations, and inferences made in this paper are straightforward, there are a couple of major concerns and several minor issues to be addressed before it is ready for publication.

First, host-association records are notoriously quite bad – in fact, some of the references included in the literature cited make this exact point and call for more field data specifically focused on trophic relationships. The methods give very little information on all of the sources, other than they are "citable" and locality and host plant data, so it is very difficult to assess the quality of the data going into these analyses. There needs to be some effort to verify or to cull out poor sources or to at least downplay them. One option would be to repeat the analyses with the best quality data (i.e. field collected using standardized methods). Related to this issue is the classic Fox and Morrow (1981) issue of local versus regional specialization (or specialization as a species trait), which is ignored by only including measures of local specialization. Furthermore, the propensity for host-shifting seems to be ignored – one example of how this latter issue could be partially addressed is by examining the proportion of exotic hosts (which were excluded from analyses for this paper) fed on by all species in the study and including propensity to shift in the simulations.

Second, much of the network complexity (and interaction diversity) of caterpillar webs is made up of rare interactions or non-nested unique feeding associations, especially when examining these webs at the local scales examined here – again, several of the references included in the literature cited make this point. Although the authors acknowledge the importance of plant diversity and associated web diversity in the final paragraph of the main text, this is really downplayed. One clear fix for this would be to conduct more thorough network analyses – the methodological approaches for ecology have exploded in recent years and include some powerful and interesting approaches (e.g., see Poisot's recent work).

For both of these issues, it would be helpful to see how the distributions and the simulation results might change at different scales and with different input parameters (see minor issues below too).

Minor issues:

Lines 26-27: This is a strange idea that caterpillars cannot avoid secondary metabolites (and perhaps

not the best reference related to this). There are abundant physiological adaptations that allow caterpillars to avoid toxic or deterrent plant compounds. The first part of the sentence is fine though.

Line 37: What does "vast majority" mean? It would be helpful to present an actual value here.

Line 43: It should be clearer in the main text that this was this compared to other distributions (as explained in lines 192-194). It also seems like it would be useful to treat distribution type as a response variable in a logit model (or something similar) to see if there are certain parameters (e.g., bioregion) that push a distribution towards one type (e.g., Pareto) versus another. Also, some details (a sentence is fine) on how "goft" works would be helpful. Finally, more sample distributions other than the one example in Figure 1 would be very helpful.

Lines 61-67: this is a pretty cursory network analysis - current approaches are a lot more sophisticated and can get to a variety of additional and important issues, such as the appropriate scale for addressing these questions.

Lines 65-67: these happen to be many of the species that are the best studied host plants in plant-caterpillar research.

Line 119-126: This concluding part is pretty important and perhaps downplayed too much. To really maintain reticulate networks or "interaction diversity," more hosts are certainly needed. The simulation should include interaction diversity as a response variable.

Lines 206-210: more details are needed on the method detailed in Harvey et al.

Lines 212-213: why not use diversity equivalents from different Hill numbers rather than just richness?

Lines 251-260: the input parameters for the simulations seem haphazard (e.g., for the first set of simulations, the total number of plants and number of keystone plants included). A sensitivity analysis would be very useful here to examine ranges of inputs.

Lee Dyer

****For ease of reading, all of our point-by-point responses to reviewer comments are in bold, and dark blue font. In our manuscript, changes are indicated with yellow highlighting****

SUMMARY

In this revision, we made substantial analytical additions and manuscript improvements. Here, we summarize our main changes (with additional details provided in the attached point-by-point response): First, in response to both reviewers, we include more detail about the network analysis and additional citations that bolster the robustness, and contemporary relevancy of this approach. While we did not include a different network approach, we are confident that the new details provide sufficient evidence that will alleviate concerns from reviewer #2.

Secondly, we address the comment from reviewer #2 about the quality of our data. In doing so, we include more detail about the sources used to create the dataset, as well as include a 223-page list of the peer-reviewed studies (n=3614) for hostplant information (Appendix 2).

In addition, to address concerns about data quality, we conducted a supplementary network analysis on a local field-collected dataset. We compare our field-derived results to our literature-derived results and show that the results are very similar with high correlation ($R=0.83$). We include this analysis in Appendix 1 for readers.

Importantly, we also added language in the discussion to highlight the value and utility of this analytical approach for local-scale, high quality datasets, as a means to encourage readers to use this design as a means to evaluate the contributions of plants for their study system of choice.

Thirdly, reviewer #1 had concerns about the influence of county size in the dataset. We include information on county size in our tables, as well as include a new analysis comparing distribution of parameter to county size and show that distributions are not strongly affected by variation in county size.

Fourthly, both reviewers brought up the issue of caterpillars that switch to new host plants, or species that use host plants that are undetected. To address this, we included a new stochastic element to our simulations by using the proportion of non-native host plants in a species' diet as a way to parameterize the probability of using a new host plant. Including this element does not change our results but makes our inference more realistic. Rather than include this as a supplemental analysis, we include this new simulation in the main text that includes host switching.

Fifthly, in response to reviewer #2 and the editor, we include a new sensitivity analysis to our simulation that addresses how Lepidoptera species richness changes with the addition of 1) total host plants, and 2) keystone host plants. After completing this revision, we felt that this was a substantial improvement to the simulations we included in the first draft (testing differences where # of plants is fixed to 20), and the graphs we created better illustrate our point. Therefore, we opted to include a new figure and results for our

simulation for both woody and herbaceous plants in the main text and deleted our simulation where plant richness was fixed.

Finally, we made all changes suggested by the reviewers with regard to language to improve clarity and content to bolster our introduction and discussion.

POINT-BY-POINT RESPONSE

Main comments:

1) The narrative of the introduction paragraph 1 lines 12-20 seems to “cherry-picked” as one of the best long-term monitoring studies of insects show increases and decreases of moths over time (Macgregor et al NEE, 2019), whereas some of the studies cited in this manuscript as evidence of unequivocal declines are commentaries that provide no data regarding declines (Eisenhauer et al Nat. Com 2019), or have highly controversial experimental designs (Hallmann et al 2017 PLoS ONE).

DLN: We appreciate the reviewer pointing out that an excellent study of Lepidoptera populations was omitted. At the time this paper was being finalized this important study had just been released but we are more than willing to include this paper now. We reworded the first sentence to highlight that we chose these ‘high-profile’ studies precisely because they have alerted the public to the importance of insect declines (Line 12). Our study is not meant as a comment on insect declines broadly, but as a solution to advance insect conservation which is now more apparent in the public eye. We also moved the commentary papers to the ones that we specify as reviews. Finally, we include the Macgregor paper both here, and on Line 16, with the following sentence (additions underlined).

“Although some species may be stable or increasing (MacGregor et al. 2019), reductions in any Lepidoptera populations are particularly concerning given the importance of these insects in terrestrial food webs; caterpillars transfer more energy from plants to other animals than all other herbivores combined (12).”

2) The key result of the manuscript is that a small percentage of plant genera support the majority of Lepidoptera. This is determined looking at presence and absence of lepidoptera on host plants, rather than abundance of lepidoptera. If abundant species are found on separate host plant species (sensu segregation-aggregation patterns reported in Calatayud et al 2019 Nature Ecology and Evolution) then a wider variety of hosts would need to be conserve total lepidopteran biomass, but a narrower range of hosts would be needed to conserve species richness. The influence of presence-absence vs abundance data on the outcome of the results should be qualitatively described as an important caveat, or quantitatively included in the simulations as a sensitivity analysis.

DLN: We agree with the reviewer that the lack of abundance data is an important caveat that we should speak to. Unfortunately, at this time, abundance data at the scale of our study is not available. In addition, for the number of Lepidoptera species included in this

study (>12,000), the relative differences in abundance (or biomass) in a given community, or among different host plants, is unavailable. Thus, out of necessity, our current analysis must be restricted to species richness. Effectively including abundance in a sensitivity analysis for our simulations without access to real data would require unfounded speculation and, in our opinion, may not be informative, given that our mode of inference throughout this paper is centered on richness. Thus, we have chosen to explain this caveat qualitatively.

To respond to this comment, we clarify the intent to preserve species richness on line 34. We also include reasons why species richness is used here over abundance or biomass (Line 38-42) and provide citations later in the text that show that richness is often strongly correlated with abundance and biomass at the community level (Line 169-172). Finally, we again highlight this caveat in our discussion in lines 155-170 and urge readers to use these methods with high-quality networks that have better information on abundance, biomass and interaction strengths.

3) The authors indicate that the host plant associations are filtered because not all lepidopterans occur in all areas where their potential host plants occur. However, the choice for units for variables are not standardized metrics (i.e. county area) or carefully justified (ie. genus). Why were plant genera chosen as a unit of conservation? How does the number of species per genera influence their importance? How did the authors account for differences in sizes of counties? Looking at the map in Figure 1B there appears to be quite a big size range. Noting that the largest county in the US is 51,950 km² and the smallest in the continental US is 59 km². It is likely then that the network of potential interactions in larger or more heterogeneous counties end up with more interactions that are not actually realized. ??

DLN: We appreciate the reviewer bringing up these important points about our dataset. We answer these questions below:

- a) *Why were plant genera chosen as a unit of conservation?* **Plant genera were included as the unit of conservation because data on caterpillar-host plant associations is most available and accurate at the genus level. We specify this reasoning in Lines 199-203.**
- b) *How does the number of species per genera influence their importance?* **The number of plant species in a genus is one hypothesis as to why some genera are disproportionately important. However, there are several other competing (and not mutually exclusive) hypotheses that may also explain these patterns, including abundance, phylogenetic isolation, foliar chemistry of a genus, genus age, and geographic range. We posit these potential mechanisms on Lines 139-141 with corresponding citations. Addressing which mechanisms primarily drives the patterns we see is outside the scope of this study which is to demonstrate that a pattern exists; however, we hope that this paper will inspire future researchers to address these research avenues with more mechanistic approaches.**

- c) *How did the authors account for differences in sizes of counties? Although the counties of the United States vary significantly in area, this study does not use data from every county. As possible, we limited the dataset to counties that were somewhat similar in size to reduce the influence of this factor on variance (e.g. no counties in Hawaii or Alaska are included). To describe the variation in counties used in this study, we include the range of county size by land area in the methods as well as the mean and SD (Line 243-246). As a new addition to the paper, we also test for effects of county size on our shape and scale parameters which are added to Table S1 and in the main text where needed.*
- d) *It is likely then that the network of potential interactions in larger or more heterogeneous counties end up with more interactions that are not actually realized. ?? Since our data are based on host plant records, it is not possible to know which interactions ascribed to a county may not be realized or be realized at low frequencies. However, we emphasize that our data represent ‘potential’ interactions, which may be dynamic due to a multitude of factors such as climate change, land use change, cultivation etc. For example, more native species are increasing their distributions northward as temperatures increase, thus the frequency of species interactions may be also increasing. Given that distribution metrics were not meaningfully related to county size, we don’t think county size has any influence on our conclusions. We hope that this approach will be used to evaluate fine-scale abundance-based local datasets in the future.*

Specific comments:

Line 53: multiple effect sizes but only one value of β ? The reader is not informed to what this parameter refers until the methods at the end. A brief description that parameter in the text here would improve ease of reading.

DLN: We include a detailed definition of parameters Θ and α prior to these results in line 50-56 prior to these results. We also changed this sentence to include “all effect sizes were negligible and close to zero” and removed the effect sizes since they are all similar and the reader can refer to the exact numbers in the table S1.

Figure S2 slopes were significant but effect sizes were near zero—How is effect size calculated?

DLN: Here we base effect size on the unstandardized coefficients, which were <0.0001 for each 1-unit change, and <0.01 for each 100-unit change. We added in “based on unstandardized coefficients” to clarify (L59).

Line 61-63- network analysis is a bit vague. Stability is not clearly defined—add more explanation in addition to the Harvey et al reference.

DLN: We added additional details for each metric used in this analysis (species richness, extinction sensitivity, network stability). We also include a definition for network stability (taken from Sauve et al. 2016 from whom the methods were borrowed) and brief description of how the metric is derived for this analysis. We direct the reader to our

Methods as well as Harvey et al. 2016 and Sauve et al. 2016 for additional analytical details (Line 64-84 & 296-315)

Line 173: The terms ‘zone’, ‘ecoregion’, and ‘bioregion’ are all used in the methods. If they refer to the same thing, they should be standardized, preferably consistent with the cited reference.

DLN: We appreciate the reviewer pointing out the inconsistency in our terminology. We changed all terms to be ‘ecoregions’ to be consistent with our cited reference.

The cited reference to ecoregions describes a hierarchy of regional partitioning. Yet the application of the hierarchy, or which level is chosen is not clearly described in this manuscript.

DLN: We added a sentence to describe which levels were used for county selection and analyses. L 229-230.

Line 199: Why is productive specified?

DLN: The word productive was superfluous here and was removed. We changed this sentence to read “We then tested for differences in the distribution of caterpillar richness among plants by county-level diversity and location metrics” L266.

Reviewer #2 (Remarks to the Author):

This interesting and important manuscript is based on a substantial trophic database that is focused on the host plants of lepidopteran larvae (as well as the consumers of these caterpillars). A clear pattern of skewed distribution of food plant genera show a small percentage of “keystone” plants, and the authors suggest that these plants are essential for restoration or for preventing continued declines of Lepidoptera, which are important pollinators and primary consumers.

The work is certainly important with broad scientific appeal and interest and it is worthy of publication in a high impact journal. While the methods, interpretations, and inferences made in this paper are straightforward, there are a couple of major concerns and several minor issues to be addressed before it is ready for publication.

DLN: We thank the reviewer for recognizing the importance of this work. We hope that our additions and explanations alleviate their concerns.

First, host-association records are notoriously quite bad – in fact, some of the references included in the literature cited make this exact point and call for more field data specifically focused on trophic relationships. The methods give very little information on all of the sources, other than they are “citable” and locality and host plant data, so it is very difficult to assess the quality of the data going into these analyses.

There needs to be some effort to verify or to cull out poor sources or to at least downplay them. One option would be to repeat the analyses with the best quality data (i.e. field collected using standardized methods). Related to this issue is the classic Fox and Morrow (1981) issue of local versus regional specialization (or specialization as a species trait), which is ignored by only including measures of local specialization.

DLN: While we recognize that host-plant records have weaknesses, particularly at a fine scale (e.g. if a caterpillar is erroneously attributed to feeding on a plant that is not a viable host), these weaknesses are relatively consistent across plant genera and thus would not conceal real patterns. Thus, we believe that the relative comparison of plant taxa at the broad scale needed for this analysis is robust to host record errors. Most important, however, is that at present, these are the best available data regarding host plant importance on a continental scale. If inferences are needed at a local scale, our paper provides a thorough analytical framework for which better quality data could be applied (included in Appendix 1). For example, interaction networks that include variation such as Lepidoptera abundance, host plant abundance, or regional host plant specialization could be useful if data are available at a local scale. We predict that local specialization *vis a vis* Fox and Morrow would only accent the importance of host plant specialists on particular plants and reduce the importance of generalists, resulting in an even more striking pattern than what we found.

We did our best to filter the quality of data sources by only using records published in peer-reviewed literature (in other words, we did not use online sources such as BugGuide.net, social media records, iNaturalist, etc.). Nevertheless, we recognize that published records suffer from plant misidentifications, a bias toward economically important plant taxa, Lepidoptera misidentifications, taxonomic revisions, and incidental use by caterpillars. But again, although they occur, we feel such errors are not frequent enough in the literature given the enormous size of the data-set to produce the magnitude of the pattern our study reveals. We include a caveat about this limitation in the discussion and discuss the usefulness of the analytical framework for better quality data in Lines 153-165.

In addition, we are including as a supplement the list of all 3500+ sources so that readers can assess exactly where the data came from (Appendix 2).

Note: we also include a supplementary analysis on a dataset that was field collected using standardized methods (data from Richard et al. 2019, Biological Invasions, included as Appendix 1). With this smaller dataset, we show that 1) distributions are qualitatively similar to the skewed distribution found with host plant data, 2) network analyses yield plants that are disproportionately important and 3) scores derived from field-collected data were strongly correlated with those derived from the host plant dataset for the nearest county (Sussex DE). For brevity, we keep this analysis in the supplement, but refer to it in the text for reference (Line 157).

Furthermore, the propensity for host-shifting seems to be ignored – one example of how this latter issue could be partially addressed is by examining the proportion of exotic hosts (which

were excluded from analyses for this paper) fed on by all species in the study and including propensity to shift in the simulations.

DLN: We thank the reviewer for raising this point and for providing a constructive solution. We agree with the reviewer that this should be included in our analyses, and we address this issue in two ways. First, we include the reviewer's suggestion by including the proportion of exotic host plants as a proxy for the propensity to shift. Specifically, we calculated the probability of shifting to a host plant as:

$$P_{ic}(\text{host plant shift}) = \frac{E_{ic}}{N_c - H_{ic}} * N$$

Where P_{ic} is the probability of host plant shifting by Lepidoptera species i in county c , E_{ic} is the proportion of exotic plants used by Lepidoptera species i in county c , N_c is the number of native host plants in county c , H_{ic} is the number of total native host plants used by Lepidoptera species i in county c , and N is the number of plants included in the simulation (from 1 to 50). The result is the probability that Lepidoptera species i in county c shifted to at least one of the N plants used in the simulation. We then use this probability to predict whether a caterpillar is included (1) or a caterpillar is not included (0) in each iteration and add these species to those included from known hosts. The updated code for this simulation, which also includes other suggestions by reviewer #2, can be found in the supplement (Appendix 3 – code).

Second, for reference, we also include a supplementary graph showing the difference in native interactions and novel interactions to illustrate that the vast majority of interactions in this dataset are native.

Second, much of the network complexity (and interaction diversity) of caterpillar webs is made up of rare interactions or non-nested unique feeding associations, especially when examining these webs at the local scales examined here – again, several of the references included in the literature cited make this point. Although the authors acknowledge the importance of plant diversity and associated web diversity in the final paragraph of the main text, this is really downplayed. One clear fix for this would be to conduct more thorough network analyses – the methodological approaches for ecology have exploded in recent years and include some powerful and interesting approaches (e.g., see Poisot's recent work).

DLN: While we agree with the reviewer that there are more complex network analyses possible, we feel that the approach we took here is both appropriate and robust, particularly in regard to the continental-wide dataset available. In response to reviewer #1 we include additional descriptions of the three-step network analysis that we believe provides better clarification of the complexity of our approach which is grounded in current network methods. (Lines 73-84, L296-315).

We recognize that much of the narrative in the paper emphasizes number of Lepidoptera species as this is a metric that is most recognizable to the general public. However, our assessment of national ‘importance’ in the context of this network analysis is based on the results from three complementary metrics. To reiterate, the network analysis includes: 1) the number of interactions each host plant supports, 2) the number of specialized species (1 host plant) that a plant supports (i.e. the extinction sensitivity) and 3) the contribution of that plant to network stability. We reviewed the text to make sure it is clear when we are talking about # of species versus the results of the network analysis.

The network stability approach used here is borrowed from several current papers in leading ecological journals in the field that have demonstrated both the utility and sophistication of this approach which we cite in the text:

- Sauve, A.M., Thébault, E., Pocock, M.J. and Fontaine, C., 2016. How plants connect pollination and herbivory networks and their contribution to community stability. *Ecology*, 97(4), pp.908-917.
- Sauve, A.M., Fontaine, C. and Thébault, E., 2014. Structure–stability relationships in networks combining mutualistic and antagonistic interactions. *Oikos*, 123(3), pp.378-384.
- Harvey, E., Gounand, I., Ward, C.L. and Altermatt, F., 2017. Bridging ecology and conservation: from ecological networks to ecosystem function. *Journal of Applied Ecology*, 54(2), pp.371-379.
- Tang, S., Pawar, S. & Allesina, S. (2014) Correlation between interaction strengths drives stability in large ecological networks. *Ecology Letters*, 17, 1094–1100.

Moreover, in Harvey et al. 2017, the authors highlight how a network approach that incorporates multiple metrics of diversity, extinction and stability could be a valuable resource for conservation managers. The paper we’ve submitted is an answer to their suggestion of how network structure can inform conservation application by using their methods to provide an explicit framework that managers could use to both inform plant selection, or adapt finer local scale plant-caterpillar datasets. We include a few new sentences in the discussion to address this value (Line 153-157).

In conclusion, we emphasize that this approach is best suited for the binary, bipartite data that characterizes our data set and is the best data currently available for our analysis at a continental scale. We also believe this multi-step approach is the ideal tradeoff between rigorous analytical power and unambiguous utility for broad audiences interested in managing for interaction diversity.

For both of these issues, it would be helpful to see how the distributions and the simulation results might change at different scales and with different input parameters (see minor issues below too).

DLN: We specifically address this concern below.

Minor issues:

Lines 26-27: This is a strange idea that caterpillars cannot avoid secondary metabolites (and perhaps not the best reference related to this). There are abundant physiological adaptations that allow caterpillars to avoid toxic or deterrent plant compounds. The first part of the sentence is fine though.

DLN: Our intent here is that caterpillars are exposed to metabolites due to their mode of feeding. Therefore, we added the qualifier, “because their feeding mode typically makes it impossible to avoid exposure to toxic or deterrent phytochemicals without physiological adaptations related to host plant specialization” to clarify. (Line 28)

Line 37: What does "vast majority" mean? It would be helpful to present an actual value here.

DLN: Because this sentence is describing just Lepidoptera diversity, we include the average % of plants per county that support >90% of Lepidoptera and table 3 which shows this (Line 61). The rest of our results expand on this through multiple complementary analyses.

Line 43: It should be clearer in the main text that this was this compared to other distributions (as explained in lines 192-194). It also seems like it would be useful to treat distribution type as a response variable in a logit model (or something similar) to see if there are certain parameters (e.g., bioregion) that push a distribution towards one type (e.g., Pareto) versus another. Also, some details (a sentence is fine) on how “goft” works would be helpful. Finally, more sample distributions other than the one example in Figure 1 would be very helpful.

DLN: We include in the main text that we compared multiple distribution types before concluding that the exponential Gamma was the best fit (Line 47). We appreciate the reviewer offering a constructive suggestion to treat distribution type as a response; however, given that only 6/83 (7%) of the counties did not fit a Gamma distribution (and did not fit our other skewed distributions), we do not think this would add valuable information to our analysis.

To clarify the functions from the ‘goft’ package, we added this sentence: “Functions in the ‘goft’ package use parametric bootstrap tests for the null hypothesis that a distribution fits a tested distribution. We tested each county (n=83) and each distribution type (n=3) separately”. (Line 262)

Finally, we include the other tested distributions in our Fig 1a example (exponential, gamma, pareto) as well as the well-known normal distribution for reference. In order to accommodate the more complex graph, we rearranged the positions of the 4 graphs and their corresponding captions (Figure 1).

Lines 61-67: this is a pretty cursory network analysis - current approaches are a lot more sophisticated and can get to a variety of additional and important issues, such as the appropriate scale for addressing these questions.

DLN: We answer this concern in the reviewer's comments above. We hope that the additional detail provides the necessary evidence that our approach is sophisticated, robust, and appropriate for this investigation.

Lines 65-67: these happen to be many of the species that are the best studied host plants in plant-caterpillar research.

DLN: We posit that these may be the best studied species precisely because they support the most caterpillar diversity, most specialized species, and are most relevant to local ecosystems. While it is possible that other plant genera house a vast diversity of Lepidoptera that have not yet been recognized, after more than a century of active collecting, we believe this to be very unlikely. We added a sentence here to address this point, as well as mention that the value of these plants is likely underappreciated in managed ecosystems where the majority of plants are selected (i.e. urban, agroforestry, etc.), which is a major take-home message of this paper. (Line 90-94)

Line 119-126: This concluding part is pretty important and perhaps downplayed too much. To really maintain reticulate networks or "interaction diversity," more hosts are certainly needed. The simulation should include interaction diversity as a response variable.

DLN: We agree with the reviewer that we should consider interaction diversity explicitly. We do this using the approach from both Dyer et al. 2010 *Biotropica* & Dell et al. 2019 *Frontiers in Ecology and Evolution* by including the alpha richness of interactions. We addressed this by adding to our simulation to include both species richness and interaction richness as two separate measures of diversity. Since abundance data are not available, interaction abundance was not included. The results of this analysis are shown in Fig 2 which shows the accumulation of both Lepidoptera richness (a) and Interaction Richness (b) as both plant genus richness and keystone plant richness increase. To make comparisons between counties, we convert these values into % Lepidoptera richness and % interactions from the total possible species/interactions (from woody and herbaceous plants separately). In both responses, we show that while increasing plant genus richness does increase Lepidoptera and interaction diversity, comparable values can be achieved with far fewer plants when keystone species are included for woody plants. In herbaceous plants, even a 3-fold increase in plant diversity does not replicate the species richness/interactions supported by keystone plants. This strongly supports our inference that increases in plant richness can increase species and interaction diversity just by the increased probability that keystone plants are included in a random pull. While plant genus richness is surely important, and supports rare interactions, including keystone plants is much more efficient and powerful. We speak to this in L173-179.

Lines 206-210: more details are needed on the method detailed in Harvey et al.

DLN: In response to reviewer #1 and #2, we include additional details about the analysis. We believe that these details will also alleviate reviewer #2's concern about the sophistication of this analysis. (Lines 73-84, L296-315).

Lines 212-213: why not use diversity equivalents from different Hill numbers rather than just richness?

DLN: Our understanding is that in order to calculate Hill numbers to include non-detected, hypothetical species, we would have to have a frequency measure (relative abundance or sampling information, etc.) for the number of times a caterpillar was observed interacting with a host plant. One option would be to include the number of pieces of literature (from the >3500) that mentions an interaction. However, as both reviewers will likely agree, host plant literature has limitations to its utility in this way. Using published interactions as a metric of sampling could be biased by plants being studied more than others, or, even more likely, interactions not being published because they are already known. Thus, in our opinion, the best way to use this dataset is to limit the response to a binary presence/absence data. In addition, our network analysis and simulation require knowledge of known species using host plants, so estimated hypothetical species could not be included.

However, we think the reviewer brings up a good point: there are some species that may use a host plant that were not accounted for. In response to reviewer #1 as well, we included a new parameter to model the possibility of host range expansion (i.e. using a new host plant) in our simulations. Although host range expansion is not the same as undetected species, we believe this addresses some of the uncertainty around the 'true' community of caterpillars that occupies a host plant by including this element of inherent stochasticity.

The host plant dataset here spans >3500 literature references from >100 years of entomological observations. While every present (or future) interaction may not be fully accounted for, we believe the relative differences among plants are sound. In lieu of more fine-scale data that include abundance/biomass/etc. (addressed in Lines 153-172), we believe using the raw richness counts is appropriate.

Lines 251-260: the input parameters for the simulations seem haphazard (**e.g., for the first set of simulations, the total number of plants and number of keystone plants included**). A sensitivity analysis would be very useful here to examine ranges of inputs.

DLN: As requested by Reviewer#2 and the Editor, we have added this suggestion to our simulations. We thank you both for this suggestion, as we think this is a valuable and important contribution to this work. Our new simulation shows how Lepidoptera richness and interaction richness increase as both plant richness and keystone plant richness increases. Because we think that this approach is actually more informative than our previous approach (our previous approach had a fixed number of plants and keystone plants) and it combines the information from both simulations into one, we have decided to

keep this as our main analysis. In other words, this is not included as a supplementary sensitivity analysis, but rather at the forefront of our analysis and inference in this manuscript.

We add figures for woody plants (figure 2) and herbaceous plants (Figure S7) which shows how Lepidoptera and interaction richness changes with a range of plant host and keystone plant host inputs. This figure clearly makes the point we emphasize in the discussion: that while increases in plant diversity always are beneficial for supporting herbivorous caterpillars, the biodiversity supported can be more effective and efficient by including keystone plants.

Reviewers' Comments:

Reviewer #1:

Remarks to the Author:

I carefully read the revised manuscript, all supplementary material and response to reviewers comments. I appreciated the first submission, but had raised some concerns about the influence of species abundance distributions, and differences in range sizes of hosts and caterpillars that could alter distribution of interactions. The concerns I raised during the first round of review have been adequately addressed, and the authors should be commended for their thoughtful manuscript. This manuscript is a nice example of moving from network theory to conservation practice.

Reviewer #2:

Remarks to the Author:

The authors did an excellent job in this revision and have adequately addressed reviewer concerns. I will repeat my earlier assessment that the work important with broad scientific appeal and interest, and it is worthy of publication in a high impact journal. I think the work could be published as is. I have a couple of general comments that may or may not be relevant and a few minor issues to point out. Both the general and minor comments could be ignored.

General comments

First, I do not really agree with the previous reviewer comment that "the narrative of the introduction paragraph 1 lines 12-20 seems to "cherry-picked" as one of the best long-term monitoring studies of insects show increases and decreases of moths over time (Macgregor et al NEE, 2019)." I am not sure how relevant the Macgregor study is to this paper, given that they focus on adult biomass in the UK, not on immatures and not on diversity, both of which are more relevant here – Salcido et al. 2020 seems more relevant (Salcido, D.M., Forister, M.L., Lopez, H.G. and Dyer, L.A., 2020. Loss of dominant caterpillar genera in a protected tropical forest. Scientific reports, 10.). I am not suggesting that Salcido et al. be cited (I am a coauthor of the paper), but I want to make it clear that the first part of the paper does not suffer (and never did) from cherry picking. Diversity declines are obvious and have been documented for a huge number of taxa. The Salcido paper mentioned above documents dramatic declines of abundance and diversity of immature Lepidoptera (rather than just biomass of adults) and also examines networks and interaction diversity as response variables. Again, the Macgregor paper actually shows recent overall declines (after an initial increase) using an impressive time series of over 1% per year since 1983, but that study does not include an assessment of diversity declines, and only includes adults. As the authors point out in their rebuttal, this is not a paper about insect biomass declines, but I do think that the results reported here are very important in the context of loss of caterpillar diversity and rapid changes to plant-caterpillar networks.

Second, I still think that some issues of how the data quality are not stated strongly enough, but I don't think it requires any editing to this version of the manuscript. My main complaint is that the missing data are underappreciated. There are countless plant-Lepidoptera associations that await to be discovered in the US (or that have been discovered but not published), and this would be clear if one examined rarefaction curves for novel interactions using quantitative caterpillar survey data from collection sites. So I don't think it is particularly useful to make the statement that "138 native woody genera and 860 native herbaceous genera are not known to support any Lepidoptera diversity" (lines 147-149). Surely most of these taxa support caterpillars, they are simply under-sampled. If you look hard enough, you will find caterpillars on plants that supposedly do not support them, and then the questions arise as to whether or not those plants support rare species, high diversities of uncommon species, unattractive caterpillar families, low densities of rare caterpillars, and many related questions. In their rebuttal, the authors state, "While it is possible that other plant genera house a vast diversity

of Lepidoptera that have not yet been recognized, after more than a century of active collecting, we believe this to be very unlikely." I do not see any evidence of this in the literature or in the field (especially if you delete the word, "vast") – there is clearly a sampling bias towards the top 5 host genera, and there is certainly a long list of genera that have almost no history of concerted caterpillar sampling. For example, one can walk out into a Pinyon-Juniper woodland anywhere in the west and find dozens of new host affiliations on Juniperus; there are countless other examples of reticulate food webs that await to be discovered – they simply need to be sampled, especially in the West.

Minor issues:

Line 99 (and elsewhere) – why not use "ratios" (and use decimals for estimates throughout) or at least spell out "percent"?

Lines 228-231: I think this is a great classification system for ecoregions – no change recommended here, I just want to mention that for future studies, you could consider the Dinerstein et al. system (Dinerstein, E., Olson, D., Joshi, A., Vynne, C., Burgess, N.D., Wikramanayake, E., Hahn, N., Palminteri, S., Hedao, P., Noss, R. and Hansen, M., 2017. An ecoregion-based approach to protecting half the terrestrial realm. *BioScience*, 67(6), pp.534-545.) that includes this excellent web resource: <http://ecoregions2017.appspot.com/>

Lines 300-304: This is not very clear as stated. Probably one more sentence to explain how this estimates some sort of stability would go a long way (something like what is already explained in lines 74-78).

Line 348: What is the justification for only 100 iterations per county? If it is just for computational efficiency, that would be easy enough to state here.

RESPONSE TO REVIEWERS' COMMENTS

Reviewer #1 (Remarks to the Author):

I carefully read the revised manuscript, all supplementary material and response to reviewers comments. I appreciated the first submission, but had raised some concerns about the influence of species abundance distributions, and differences in range sizes of hosts and caterpillars that could alter distribution of interactions. The concerns I raised during the first round of review have been adequately addressed, and the authors should be commended for their thoughtful manuscript. This manuscript is a nice example of moving from network theory to conservation practice.

DLN: We appreciate reviewer #1 taking the time to reread our revised manuscript.

Reviewer #2 (Remarks to the Author):

The authors did an excellent job in this revision and have adequately addressed reviewer concerns. I will repeat my earlier assessment that the work important with broad scientific appeal and interest, and it is worthy of publication in a high impact journal. I think the work could be published as is. I have a couple of general comments that may or may not be relevant and a few minor issues to point out. Both the general and minor comments could be ignored.

DLN: We appreciate reviewer #2 taking the time to reread our revised manuscript and for including their helpful and insightful comments.

General comments

First, I do not really agree with the previous reviewer comment that “the narrative of the introduction paragraph 1 lines 12-20 seems to “cherry-picked” as one of the best long-term monitoring studies of insects show increases and decreases of moths over time (Macgregor et al NEE, 2019).” I am not sure how relevant the Macgregor study is to this paper, given that they focus on adult biomass in the UK, not on immatures and not on diversity, both of which are more relevant here – Salcido et al. 2020 seems more relevant (Salcido, D.M., Forister, M.L., Lopez, H.G. and Dyer, L.A., 2020. Loss of dominant caterpillar genera in a protected tropical forest. Scientific reports, 10.). I am not suggesting that Salcido et al. be cited (I am a coauthor of the paper), but I want to make it clear that the first part of the paper does not suffer (and never did) from cherry picking. Diversity declines are obvious and have been documented for a huge number of taxa. The Salcido paper mentioned above documents dramatic declines of abundance and diversity of immature Lepidoptera (rather than just biomass of adults) and also examines networks and interaction diversity as response variables. Again, the Macgregor paper actually shows recent overall declines (after an initial increase) using an impressive time series of over 1% per year since 1983, but that study does not include an assessment of diversity declines, and only includes adults. As the authors point out in their rebuttal, this is not a paper about insect biomass declines, but I do think that the results reported here are very important in the context of loss of caterpillar diversity and rapid changes to plant-caterpillar networks.

DLN: We appreciate reviewer #1 sharing this new paper with us and their support. We are not including the Salcido et al. paper it at this stage in the manuscript but look forward to citing it in our future work as it looks like a very important contribution. The purpose of the beginning of

this paragraph was to frame our study as a response to recent evidence of insect declines which we feel is currently adequate. We opted to include several papers across taxa and do not rely on just studies of caterpillars or even Lepidoptera. We also think that the MacGregor paper is still a good example to include because it shows that biomass declined even when there was high variation between species in the long term response.

Second, I still think that some issues of how the data quality are not stated strongly enough, but I don't think it requires any editing to this version of the manuscript. My main complaint is that the missing data are underappreciated. There are countless plant-Lepidoptera associations that await to be discovered in the US (or that have been discovered but not published), and this would be clear if one examined rarefaction curves for novel interactions using quantitative caterpillar survey data from collection sites. So I don't think it is particularly useful to make the statement that "138 native woody genera and 860 native herbaceous genera are not known to support any Lepidoptera diversity" (lines 147-149). Surely most of these taxa support caterpillars, they are simply under-sampled. If you look hard enough, you will find caterpillars on plants that supposedly do not support them, and then the questions arise as to whether or not those plants support rare species, high diversities of uncommon species, unattractive caterpillar families, low densities of rare caterpillars, and many related questions. In their rebuttal, the authors state, "While it is possible that other plant genera house a vast diversity of Lepidoptera that have not yet been recognized, after more than a century of active collecting, we believe this to be very unlikely." I do not see any evidence of this in the literature or in the field (especially if you delete the word, "vast") – there is clearly a sampling bias towards the top 5 host genera, and there is certainly a long list of genera that have almost no history of concerted caterpillar sampling. For example, one can walk out into a Pinyon-Juniper woodland anywhere in the west and find dozens of new host affiliations on Juniperus; there are countless other examples of reticulate food webs that await to be discovered – they simply need to be sampled, especially in the West.

DLN: We acknowledge the concern that the reviewer raises and agree that there are many interactions that are unknown due to under-sampling, particularly in understudied regions in the west and southwest. While the specifics of these interactions may be informing at local scales, we do not think that increased sampling will change the overall conclusion of this paper: That interaction networks are skewed toward few plants supporting the majority of species. To reiterate our message from our discussion, we hope that this study inspires more work at local scales to describe and quantify these networks to better refine the species that contributing disproportionately to food webs.

Although edits were not requested, we added this line to L175-178 in response to their comment. "Although future sampling will undoubtedly yield previously undocumented interactions in many of these plant genera, particularly for rare or uncommon Lepidoptera or plant species, they are unlikely to be numerous enough to substantially change patterns of distribution we report here."

We are excited about the prospect of new collaborations in the future to collect Lepidopteran interaction data across bioregions to explore the keystone plant concept further.

Minor issues:

Line 99 (and elsewhere) – why not use “ratios” (and use decimals for estimates throughout) or at least spell out “percent”?

DLN: We spelled out percent here and elsewhere.

Lines 228-231: I think this is a great classification system for ecoregions – no change recommended here, I just want to mention that for future studies, you could consider the Dinerstein et al. system (Dinerstein, E., Olson, D., Joshi, A., Vynne, C., Burgess, N.D., Wikramanayake, E., Hahn, N., Palminteri, S., Hedao, P., Noss, R. and Hansen, M., 2017. An ecoregion-based approach to protecting half the terrestrial realm. *BioScience*, 67(6), pp.534-545.) that includes this excellent web resource: <http://ecoregions2017.appspot.com/>

DLN: We appreciate this great resource! Thank you!

Lines 300-304: This is not very clear as stated. Probably one more sentence to explain how this estimates some sort of stability would go a long way (something like what is already explained in lines 74-78).

DLN: We added another sentence to improve clarity about the direction of this metric L338-L342.

Line 348: What is the justification for only 100 iterations per county? If it is just for computational efficiency, that would be easy enough to state here.

DLN: Yes, our reasoning for the low iterations is to improve efficiency since we are only computing medians across the counties. We added a line to say this on L436.